



# Impact of Dry Intrusion Events on Composition and Mixing State of Particles During Winter ACE-ENA Study

Jay M. Tomlin[1], Kevin A. Jankowski[1], Daniel P. Veghte[3,4], Swarup China[4], Peiwen Wang[5],
Matthew Fraund[6], Johannes Weis[6], Guangjie Zheng[7,8], Yang Wang[8,9], Felipe Rivera-Adorno[1],
Shira Raveh-Rubin[10], Daniel A. Knopf[5], Jian Wang[7,8], Mary K. Gilles[6], Ryan C. Moffet[11],
Alexander Laskin[1,2*]

[1]Department of Chemistry, [2]Department of Earth Atmospheric and Planetary Sciences, Purdue University, West Lafayette, IN
47907, USA
[3]Center for Electron Microscopy and Analysis, Ohio State University, Columbus, OH 43212, USA
[4]Environmental Molecular Sciences Laboratory, Pacific Northwest National Laboratory, Richland, WA 99354, USA
[5]School of Marine and Atmospheric Sciences, Stony Brook University, Stony Brook, NY 11794, USA
[6]Chemical Sciences Division, Lawrence Berkeley National Laboratory, Berkeley, CA 94720, USA
[7]Center for Aerosol Science and Engineering, Department of Energy, Environmental and Chemical Engineering, Washington
University in St. Louis, St. Louis, MO 63130, USA
[8]Environmental and Climate Science Department, Brookhaven National Laboratory, Upton, NY 11973, USA
[9]Department of Civil, Architectural and Environmental Engineering, Missouri University of Science and Technology, Rolla, MO
65409, USA
[10]Department of Earth and Planetary Sciences, Weizmann Institute of Science, Rehovot 76100, Israel
[11]Sonoma Technology, Inc., Petaluma, CA 94954, USA

*Correspondence to*: Alexander Laskin (alaskin@purdue.edu)

**Abstract.** Long-range transport of continental emission has far reaching influence over remote regions resulting in substantial
change in the size, morphology, and composition of the local aerosol population and cloud condensation nuclei (CCN) budget.
Here, we investigate the physiochemical properties of atmospheric particles collected onboard a research aircraft flown over the
Azores during the winter 2018 Aerosol and Cloud Experiment in the Eastern North Atlantic (ACE-ENA) campaign. Particles were
collected within the marine boundary layer (MBL) and free troposphere (FT), after long-range atmospheric transport episodes
facilitated by dry intrusion (DI) events. Chemical and physical properties of individual particles were investigated using
complementary capabilities of computer-controlled scanning electron microscopy and X-ray spectro-microscopy to probe particle
external and internal mixing state characteristics in the context of real-time measurements of aerosol size distribution, cloud
condensation nuclei (CCN) concentration, and back trajectory calculations. While carbonaceous particles were found to be the
dominant particle-type in the region, changes in the percent contribution of organics across the particle population (i.e., external
mixing) shifted from 68% to 43% in the MBL and from 92% to 46% in FT samples during DI events. This change in carbonaceous
contribution is counterbalanced by the increase of inorganics from 32% to 57% in the MBL and 8% to 55% in FT. The
quantification of organic volume fraction (OVF) of individual particles derived from X-ray spectro-microscopy, which relates to
the multi-component internal composition of individual particles, showed a factor of 2.06±0.16 and 1.11±0.04 increase in the MBL
and FT, respectively, among DI samples. We show that supplying particle OVF into the κ-Köhler equation can be used as a good
approximation of field measured in-situ CCN concentrations. We also report changes in the κ values between $\kappa_{MBL, \text{ non-DI}} = 0.48$ to
$\kappa_{MBL, \text{ DI}} = 0.41$ and $\kappa_{FT, \text{ non-DI}} = 0.36$ to $\kappa_{FT, \text{ DI}} = 0.33$, which is consistent with enhancements in OVF followed by the DI episodes.
Our observations suggest that entrainment of particles from long-range continental sources alters the mixing state population and
CCN properties of aerosol in the region. The work presented here provides field observation data that can inform atmospheric
models that simulate sources and particle composition in the Eastern North Atlantic.



## 1 Introduction

Marine low clouds play significant role in the world's climate and energy balance (Wood et al., 2015). They are the major factor
in increasing the Earth's albedo − fraction of solar energy reflected back into space leading to an overall cooling effect (Wood, 2012; Wood et al., 2015). Marine low clouds represent one of the leading sources of uncertainty in atmospheric models due to limited observational data, insufficient understanding of the microphysical changes that regulate these clouds, and the lack in fine model resolution to account for such processes (Bony, 2005; Klein et al., 2013). Other relevant boundary layer processes also contribute to the challenges in assessing marine low clouds such as turbulent mixing, entrainment, and emissions of aerosols and
their precursors (Pincus and Baker, 1994; Ackerman et al., 2004). In particular, the response of low altitude clouds is sensitive to aerosol perturbations, which requires a greater understanding on the processes that govern regional aerosol budget and source attribution (Levin and Cotton, 2009; Altaratz et al., 2014; Rosenfeld et al., 2019; Zheng et al., 2018, 2021). Source dependent particle size and composition can lead to changes in the clouds albedo and precipitation due to their varying efficiency to act as cloud condensation nuclei (CCN) and ice nucleating particles (INP) (Johnson et al., 2004; Hamilton et al., 2014; Zheng et al.,
2020a).

Atmospheric particles exhibit complex internal heterogeneity (Murphy and Thomson, 1997; Prather et al., 2008; Laskin et al., 2019). These particles can come from direct emissions (i.e., primary particles), or from gas-particle conversion in atmospheric reactions (i.e., secondary particles) (Reddington et al., 2011). Primary particles with complex composition include primary organic aerosols, elemental carbon (i.e., black carbon/soot), inorganic species from combustion and biomass burning sources (Toner et al.,
2006; Souri et al., 2017), and sea spray aerosol with organic components influenced by ocean biological activity (Prather et al., 2013; Pham et al., 2017). On the other hand, secondary organic aerosol (SOA) is formed from the oxidation products of volatile organic compounds (VOCs) of either biogenic or anthropogenic origin. Secondary fine particles of nitrate and sulfate are similarly formed from the oxidation of their inorganic gaseous precursors $NO_x$ and $SO_2$, respectively (National Research Council (U.S.), 2002). In marine areas, formation of sulfate aerosol is further influenced by gas-phase emissions of dimethyl sulfide (DMS) from
biota, which upon oxidation yield low volatility products such as sulfuric acid ($H_2SO_4$) (Kulmala et al., 2000) and methylsulfonic acid (MSA) (Andreae et al., 1985; Hodshire et al., 2019). Physical and chemical characteristics of individual particles such as morphology, chemical composition, hygroscopicity, lifetime, and chemical mixing state have a profound effect on their CCN activity (Cruz and Pandis, 1997; VanReken, 2003; King et al., 2012; Schmale et al., 2017; Riemer et al., 2019). Note that the term "chemical mixing state" refers to how various chemical species are mixed within individual particles (Riemer et al., 2019). The
chemical mixing state depends on emission sources and atmospheric ageing events which include, but are not limited to, biomass burning influence (Levin et al., 2010), anthropogenic emissions (Jacobson, 2001), large continental dust events (Fraund et al., 2017; Adachi et al., 2020). For example, previous studies found that within a few hours urban non-hygroscopic aerosol (i.e., mixed organic and black carbon aerosol) can accumulate a sufficient coating of hygroscopic sulfates and nitrates to increase their hygroscopicity parameter *(κ)* (Petters and Kreidenweis, 2007) from 0 to 0.1 (Wang et al., 2010).

The variability within individual atmospheric particles has been well documented by both model and field measurements across different regions worldwide such as urban (Wang et al., 2010; Ault et al., 2010, 2012; Wang et al., 2012; Fraund et al., 2017; Ren et al., 2018), rural (Vakkari et al., 2018; Tomlin et al., 2020), remote forested areas (Bondy et al., 2018), Arctic (Gunsch et al., 2017; Gonçalves et al., 2021), and marine (Ault et al., 2013; Zheng et al., 2020a, b). Long-range transport and meteorological processes such as dry intrusions (DI) and vertical mixing of air also play a significant role in the continuous evolution of particle
composition in the atmosphere (Raes, 1995; Pratt and Prather, 2010; Cubison et al., 2011; Igel et al., 2017; Zheng et al., 2020b). DI are events of dry, slantwise descending airflow from the upper troposphere in midlatitudes down through the boundary layer at





lower latitudes (Raveh-Rubin, 2017). Such intrusions of dry air, typically peaking in winter, occur with the passage of extratropical cyclones and their trailing cold fronts, i.e., in the post cold-frontal region (Wernli, 1997; Browning, 1997; Catto and Raveh-Rubin, 2019). Events of DI are strongly coupled to the boundary layer, which cools and deepens during DI, and were shown to induce

enhanced ocean heat fluxes (Raveh-Rubin and Catto, 2019; Ilotoviz et al., 2021). DI events are of particular interests as they can contain air mass with a complex distribution of aged particles having drastically different size, morphology, and composition compared to local regional aerosols, leading to changes in the local aerosol-cloud interactions and cloud lifetimes (Zheng et al., 2020b; Wang et al., 2020). For example, it has been shown that the CCN population in the remote marine boundary layer (MBL) of Eastern North Atlantic can be influenced by long-range transport of wildfire aerosols originating from North America (Zheng

et al., 2020b; Wang et al., 2021b). The properties of these wildfire aerosols facilitated by long-range transport processes alters as it undergoes aging (e.g. multiphase particle chemistry, photo-bleaching and gas-particle partitioning of organics), resulting in changes in both the optical properties and the cloud-forming potential (Jacobson, 2001; Levin et al., 2010; Zheng et al., 2020b). In particular, aged wildfire aerosol is typically dominated by accumulation mode particles, which readily serve as CCN in the region despite a substantially lower $\kappa$ value (i.e., 0.2 to 0.4) than regional highly hygroscopic aerosol of marine origin (e.g. sea spray

aerosol, $\kappa = 1.1$) (Zieger et al., 2017; Zheng et al., 2020b). Lastly, long-range transported and atmospherically aged free tropospheric particles can contribute to the ice-nucleating particle population and potentially impact cloud formation (China et al., 2017).

This paper investigates the physiochemical properties of atmospheric particles during the Aerosol and Cloud Experiment in the Eastern North Atlantic (ACE-ENA) field campaign conducted at the Azores in January-February 2018. Aircraft measurements and

onboard sampling of particles (followed by laboratory-based particle analysis) were utilized to characterize the difference in the contributions of various sources to FT and MBL aerosols under representative synoptic conditions (i.e., DI vs. non-DI periods) in this geographical area. Particle analysis included particle-type classification with statistical depth provided by computer-controlled scanning electron microscopy and a subset of particles were sampled by X-ray specto-microscopy to characterize particle chemical mixing state (internal heterogeneity). The particle-type composition, chemical mixing state, and morphology from analyzed periods

were then combined with real-time measurement of aerosol size distribution, CCN concentration, and back trajectory calculations to obtain the representative composition of particles present in the MBL during the DI events and entrainment of particles originated from North America. The data presented here provides observational input for atmospheric process models to simulate sources and particle composition in the broader North Atlantic region.

## 2 Experimental Methods

**2.1 Field Campaign and Meteorological Conditions**

Samples of atmospheric particles were collected aboard the U.S. Department of Energy Gulfstream aircraft (G-1). Flight patterns were flown between Terceira Island (38° 45' 43" N, 27° 5' 27" W) and Graciosa Island (39° 3' 12" N, 28° 7' 26" W), Portugal and within 20–30 km of Graciosa Island (Wang et al., 2021a). Flight plans were based on the projected meteorological conditions from various global forecast modes including Monitoring Atmospheric Composition and Climate, Global Forecast System, and

European Centre for Medium-Range Weather Forecasts (ECMWF). A subset of collected samples was selected for analysis based on synoptic conditions (identifying DI vs. non-DI periods) and altitudes (clear MBL and FT layers) for each day. Samples analyzed were collected during the second Intensive Operation Period of ACE-ENA, on the dates of 2018-01-19, 2018-01-21, 2018-01-24, 2018-01-25, 2018-01-26, 2018-01-28, 2018-01-30, 2018-02-01, 2018-02-08, 2018-02-11, 2018-02-15, 2018-02-16, 2018-02-19. These dates were selected due to unique transport episodes that associated with the sampling periods. DI days we identified





objectively using the Lagrangian analysis tool (LAGRANTO) version 2.0 (Sprenger and Wernli, 2015) and wind field data
obtained from the ECMWF interim reanalysis (ERA-Interim) with available 6-hourly, interpolated to 1°x1° horizontal grid
resolution, at 60 vertical hybrid levels (Dee et al., 2011). DI were identified by a systematic calculation of forward trajectories at
altitudes higher than 600 hPa, while the DI trajectories were identified based on the vertical descent of the airmasses. For a
trajectory to be termed a DI, their pressure must increase (i.e., descend in altitude) by at least 400 hPa in 48 hrs (Raveh-Rubin,
2017). If such a DI trajectory is found within a 3-degree radius circle around Graciosa, the date is considered as 'DI'. In addition,
backward trajectories for each sampling period were calculated for the end points at relevant flight altitudes (Figure S1 and S2)
using the Hybrid Single-Particle Lagrangian Integrated Trajectory (HYSPLIT) model (Stein et al., 2015; Rolph et al., 2017).
Atmospheric data from ERA Interim are analyzed additionally for the atmospheric column at Graciosa, namely, potential
temperature, equivalent potential temperature, potential vorticity and boundary-layer height. The latter is diagnosed in ERA Interim
using the critical bulk Richardson number, upon its first passing of the threshold 0.25, when scanning from the surface upwards
(ECMWF, 2007).

### 2.2 Particle Collection and In-Situ Measurements of Particle and Cloud Properties

The G-1 aircraft is equipped with sensor modules to deliver precise real-time inertial measurement, GPS, meteorological, and
turbulence data such as position, altitude, temperature, pressure relative humidity, and three-dimensional winds. For particle
collection, the G-1 was equipped with an isokinetic aerosol inlet, from which ambient aerosol was transported to individual
instruments. Particle samples were collected using a custom built time-resolved aerosol collector (TRAC) that autonomously
collected particles on substrates at preset time intervals (Laskin et al., 2006). The TRAC is a single stage impactor with an
aerodynamic cutoff size ($D_{50\%}$) of 0.36 µm (Laskin et al., 2003) coupled to a rotating disk that can hold up to 160 samples. The
disk was preloaded with microscopy substrates (Carbon Type-B film coated 400 mesh copper grids, Ted Pella, Inc.). The sampling
was performed at a single spot on the center of each substrate for 7–10 min, depending upon the flight. After each flight, sample
discs were sealed and refrigerated prior to transport for off-line multi-modal microscopy analysis.

Online measurements of aerosols abroad the G-1 include a passive cavity aerosol spectrometer-100X probe (PCASP, $D_p$ = 0.1–3.0
µm, 1 Hz resolution) and a fast integrated mobility spectrometer (FIMS, $D_p$ = 0.01–0.5 µm, 1 Hz resolution), which provided size
distributions and concentrations of ambient particles (Kulkarni and Wang, 2006; Wang et al., 2018). During all research flights, a
Nafion dryer reduced the relative humidity of the air stream in the sampling line. A CCN counter (Droplet Measurement
Technologies) measured the concentration of particles that activate at a supersaturation of 0.14%. A high-resolution time-of-flight
aerosol mass spectrometer (HR-ToF-AMS) was deployed onboard to characterize bulk non-refractory aerosol composition (i.e.,
organics, sulfate, ammonium, and chlorine) (DeCarlo et al., 2006; Zawadowicz et al., 2021). The particle size distributions and
CCN concentrations were analyzed when the liquid water content was below 0.001 g/m$^3$ to avoid periods when cloud shattering
artifacts could influence the sampled particles (Korolev et al., 2011). The liquid water content was obtained by integrating the
droplet size distributions measured by a fast cloud droplet probe (FCDP; Droplet Measurement Technologies).

Additional information on the sampling conditions is presented in the Table S1 of the supplemental file and incudes sampling
time/date, average sampling altitude, boundary layer height, particle concentration, and wind speed. The boundary layer height
was calculated based on potential temperature measurements collected for each flight. The boundary layer is limited by a well-
defined temperature inversion resulting in a maximum value of the temperature gradient as a function of height (Stull, 1988). A
summary of each flight (altitude and aerosol particle concentration vs. time) with the collection times highlighted is shown in



Figures S3 and S4. Guided by meteorological analysis and wind field data to identify DI periods, we performed offline microscopy analysis of collected particle samples across different atmospheric layers and transport episodes during the ACE-ENA campaign.

### 2.3 Methods of Particle Analysis

Morphology and elemental analysis of individual particles was performed using computer-controlled scanning electron microscopy coupled with energy dispersive X-ray spectroscopy operated at 20 kV (CCSEM/EDX; FEI Quanta 3D, EDAX Genesis). During CCSEM/EDX analysis particle samples were systematically imaged and particles larger than 100 nm are recognized, followed by automated acquisition of their individual EDX spectra (Laskin et al., 2005). EDX particle spectra with sufficient X-ray counting statistics (40–1500 photons/s) were then processed to quantify relative atomic fractions of 15 elements: C, N, O, Na, Mg, Al, Si,

P, S, Cl, K, Ca, Mn, Fe, and Cu. The EDX peak of Cu is heavily influenced by a background signal from the copper TEM grid and the sample holder made of beryllium-copper alloy. Therefore, quantified atomic fractions of Cu were excluded from particle-type classification of the analyzed particles. Two independent methods were employed for the particle-type grouping and classification: (1) *k*-means clustering and (2) rule-based particle classification. The *k*-means clustering is an unsupervised machine learning algorithm designed to group similar data sets without user intervention (Rebotier and Prather, 2007; Moffet et al., 2012). The

second approach for the categorization of particles utilizes a series of user-defined rules to separate analyzed particles into groups of typical elemental contribution (Laskin et al., 2012). For this work, the *k*-means clustering was used as a primary method for particle-type classification while the rule-based approach was used as a complementary method to build confidence on the identification of different particle-types. Details of the classification schemes are provided in the Supporting Information (Figures S5 and S6) and in previous works (Moffet et al., 2012; Tomlin et al., 2020).

Scanning transmission X-ray microscopy with near edge X-ray absorption fine structure (STXM/NEXAFS) spectroscopy was used to elucidate the chemical mixing state of individual particles based on the NEXAFS spectral data acquired at the Carbon K-edge (278–320 eV) (Hopkins et al., 2007; Moffet et al., 2010b, c). The STXM/NEXAFS was performed at the synchrotron facilities on beamlines 11.0.2.2 and 5.3.2.2 in the Advance Light Source, Lawrence Berkeley National Laboratory and on beamline 10ID-1 in the University of Saskatchewan, Canadian Light Source. STXM instrument operation is similar in both locations as described

elsewhere (Kilcoyne et al., 2003). Briefly, a set of raster scan STXM images at each of the pre-set energy levels was acquired from a synchrotron monochromated incident light focused on the sample using a Fresnel zone plate. The transmitted light is detected at each of the energy settings, and spectra of individual particles could then be reconstructed based on the Beer-Lambert law from the intensity of transmitted light over the projection area of particles compared to the particle-free regions. The recorded intensity at each energy setting ($E$) across individual pixels were converted into optical density ($OD_E$) as follows:

$$OD_E = -\ln\left(\frac{I(E)}{I_0(E)}\right) = \mu\rho t \tag{1}$$

where $I(E)$ is the intensity of light transmitted through a particle, $I_0(E)$ is the intensity of incoming light (determined as intensity of light in the particle-free areas), $\mu$ is the mass absorption coefficient, $\rho$ corresponds to the density, and $t$ is the thickness of a particle. Sequences of STXM images are acquired at closely spaced energies of $I_0(E)$ to record a "stack" of images. Then, NEXAFS spectra from individual pixels of detected particles are extracted from the stack (~96 energies over 278 to 320 eV range, 30–35 nm spatial resolution, 1 ms dwell time).


In addition, faster acquisition of STXM images at four key energies of 278 eV (pre-edge), 285.4 eV (C=C), 288.5 eV (-COOH), and 320 eV (post-edge) (15x15 μm, 30–35 nm spatial resolution, 1 ms dwell time) was employed to construct "maps" of individual particles using image processing methods reported in our earlier studies (Moffet et al., 2010a, 2013, 2016; Fraund et al., 2017).





Briefly, a series of thresholds were used to identify the mapping components including "inorganics" (IN), "organic carbon" (OC),
and "soot/elemental carbon" (EC). The total carbon (TC) was calculated as the difference between the carbon post-edge and pre-
edge OD (TC = $OD_{320eV} - OD_{278eV}$). "IN" rich regions were defined with pixels having an $OD_{278eV}$ / $OD_{320eV}$ ratio greater than 0.5.
"OC" regions are those with the abundant features corresponding to carboxylic acid functional group (-COOH), defined by the
difference between intensity of the -COOH peak and carbon pre-edge peak greater than 0 (i.e., $OD_{288.5eV} - OD_{278eV} > 0$). Finally,
EC areas are identified by comparing the value of the sp2/total carbon to that of highly oriented pyrolytic graphite (HOPG)
according to: $(OD_{285.4eV}/TC)*(OD_{HOPG, TC}/OD_{HOPG, C=C}) > 0.35$, which indicates extensive $sp^2$ bonding of carbon corresponding to
graphitic-like components (Hopkins et al., 2007).

## 3 Results and Discussion

### 3.1 Identification of dry intrusion periods

Research flights were conducted under different synoptic conditions to allow for the characterization of common aerosols, trace
gases, clouds, and precipitation. Figure 1A illustrates the typical flight pattern of the G-1 aircraft which includes multiple legs at
different altitudes, while maneuvering perpendicular and along the wind direction. These patterns allowed for the full profile of
aerosol and cloud layer along the MBL and lower FT altitudes. Figure 1B shows daily time series between 2018-01-01 to 2018-
02-28 in relation to DI events identified from ERA-Interim reanalysis. The marked black dots indicate DI air masses within a 3°
radius from 39N°, 28W° (i.e., the ENA site). The high frequency of the black dots (i.e., vertical distribution) indicates an increase
in trajectories that satisfy the DI criterion at different pressure altitudes. For example, on 2018-01-24 in Figure 1B, we see a series
of DI air parcels (black dots) at different pressure altitudes ranging from 611 m to 2360 m MSL and found to be below/above the
boundary layer as indicated by the dashed red line. Guided by the frequency of the DI air masses, we selected a subset of the time-
tagged particle samples for analysis by the complementary microscopy techniques as summarized in Table S1. To evaluate the
consistency of the sources and long-range transport trajectories, we calculated back trajectories using the HYSPLIT model (Stein
et al., 2015; Rolph et al., 2017). Figure 1C shows results of a representative HYSPLIT 72 hrs back trajectory calculations for the
research flight on 2018-01-24, which identifies long-range transport an airmass originating from North America. Trajectories were
calculated every 6 hrs from 1300 UTC 2018-01-24 to 1200 UTC 2018-01-22 at 3 starting altitudes: 100 m, 2000 m, and 3000 m.
This process was repeated with the same HYSPLIT input meteorological parameters for all research flights utilized in this work,
as shown in Figures S1 and S2.

### 3.2 Particle-type classification

A total of 38 particle sample grids from 13 (out of 19) research flights were analyzed. First, CCSEM/EDX analysis was carried
out to characterize the particle-type composition typical of different synoptic scenarios. Figure 2 shows the results of the size-
segregared particle-type population (right column) obtained from *k*-means clustering analysis of ~36,400 individual particles with
SEM images of a representative subset of particles (left column), separated between MBL versus FT flight altitudes and between
synoptic conditions of DI and non-DI sampling periods. The onboard FIMS instrument measurement provided particle size
distribution data in a range of 0.01–0.5 μm. By superimposing the CCSEM/EDX particle analysis data with the FIMS size
distribution data, we can approximate the representative composition and number concentration of potentially CCN active particles
(>0.1 μm) in the MBL and FT, during non-DI and DI periods, respectively. Note that the error bars in the particle number
concentration indicate variation in the particle size distribution values averaged across different days and synoptic conditions.



The *k*-means clustering identified 4 key clusters that we termed as: "Carbonaceous", "Ammonium Nitrates/Sulfates", "Mixed Sea Salt" and "Aged Sea Salt". While "Carbonaceous" is the dominant particle-type found across all samples, we also see that the elemental composition of each cluster (Figure S5) has common contributions from C and O elements, suggesting substantial carbon content in all existing particles, likely because of condensation of organic content and coagulation of particles. Organic aerosol in the remote MBL has been suggested to originate from VOCs such as isoprenes, monoterpenes, formic acid, nitrogenated, and

aliphatic organics released from biological activities near the sea surface, which undergo oxidation reactions leading to SOA formation (Facchini et al., 2008; Dall'Osto et al., 2012; Mungall et al., 2017). First, we compared the change in particle-type population among samples in the MBL during non-DI and DI periods. The fraction of "Carbonaceous" particles within the MBL contributed around 68% in non-DI samples and decreased to 43% in DI samples. The lower fraction of "Carbonaceous" particles during DI periods is counterbalanced by the increased of "Inorganics" shifting from 32% (non-DI periods) to 57% (DI periods).

Here, "Inorganics" is defined as the sum of "Mixed Sea Salt" (4%) + "Aged Sea Salt" (20%) + "Ammonium Nitrate/Sulfate" (33%). Shifting focus to the comparison of FT samples during non-DI and DI periods, we found that background "Carbonaceous" particles contribute to around 92% (non-DI periods) and decrerases to 46% (DI periods). Similar to MBL observations, the shift in "Carbonaceous" contribution can be attributed to an increase in "Inorganic" influence during DI events changing from 8% (non-DI periods) to 55% (DI periods). We observed that most of the "Inorganic" influence is originating from "Ammonium

Nitrate/Sulfate" contributing between 32–33% during DI periods regardless of sampling altitude (MBL vs. FT). Both "Carbonaceous" particles and "Ammonium Nitrate/Sulfate" can originate from ocean biological activity or anthropogenic sources. Typically, over marine areas, sulfate aerosol forms from oxidation of dimethyl sulfide (DMS), a common gas species emitted by biota. Sulfates are major components of accumulation mode particles in the remote marine environment (Sanchez et al., 2018; Korhonen et al., 2008). Nitrate in marine particles can also come from vertical mixing in the ocean that surges nitrate-rich deep

waters to the surface, followed by the aerosolization through wave motion (Zehr and Ward, 2002). However, the elevated contribution of "Ammonium Nitrates/Sulfates" during the DI periods suggests likely influence from anthropogenic emissions originating from North America. Inorganic aerosols such as "Ammonium Nitrates/Sulfates" are predominantly formed from the condensation of atmospheric precursors such as $SO_2$, $NH_3$, $HO_x$, and $NO_x$, which are common components of biomass burning emissions, urban areas, and agriculture activities among others (Reff et al., 2009). A study utilizing regional chemical models have

found that the mass enhancements in inorganic aerosol can reach 23% of carbonaceous enhancements as biomass burning processes accelerate secondary formation of inorganic aerosols (Souri et al., 2017). Uptake of S- and N-containing acidic species, as well as soluble organic acids, onto the preexisting sea salt particles modifies their composition through acid-displacement reactions that can be expressed in a general form of (Finlayson-Pitts, 2003; Laskin et al., 2012):

$$*NaCl_{(aq)} + HA_{(aq,\ g)} \rightarrow *NaA_{(aq)} + HCl_{(aq,g)} \tag{2}$$

where *NaCl denotes seasalt, HA represents atmospheric water-soluble acids (e.g., $HNO_3$, $H_2SO_4$, $CH_3SO_3H$ and carboxylic acids). These reactions release volatile $HCl_{(g)}$ product, leaving particles depleted in chloride and enriched in corresponding $HA_{(aq)}$ salts. Related to this acid-displacement chemistry "Mixed Sea Salt" and "Aged Sea Salt" particle-types were identified by the *k*-clustering analysis as illustrated in Figure S5. The "Mixed Sea Salt" particles contain key components of seawater (i.e., Na, Mg, and Cl; atomic fractions of Na and Cl >10% with characteristic ratio of Cl/Na ~0.6) and minor fractions (<2%) of additional elements (e.g.,

Ca, Mn, Fe, Al, and Si) suggesting internal mixing of relatively fresh sea salt with other inorganic components without extensive chloride depletion. The other cluster of "Aged Sea Salt" particles shows significant fractions of Na (~10%), but with substantially lower ratios of Cl/Na<0.1 which indicates chloride depletion (Figure S5) due to atmospheric aging. Atomic fractions of C and N elements in this type of particles are much higher than those in the "Mixed Sea Salt" cluster, while the fraction of S is much smaller.



These observations suggest that in this geographical region acid-displacement reactions in the "Aged Sea Salt" particles are mostly
driven by water-soluble carboxylic acids (common components of SOA) (Laskin et al., 2012) and nitric acid (Finlayson-Pitts,
2003), while contributions by sulfonic or sulfuric acids are minor during the wintertime. Based on the $k$-means clustering, fractions
of "Mixed Sea Salt" range from 0.5 to 3% while fractions of "Aged Sea Salt" are overall more populous and range between 0.1
and 20% across all investigated samples.

To better discriminate particle-type groups according to their composition and the acid-displacement chemistry identified through
the $k$-means clustering, a supplmental rule-based classification was performed using previously published definitions of particle-
type classes common in marine environments (Laskin et al., 2012; Tomlin et al., 2020). Results of the particle-type characterization
utilizing the rule-based assessment of their elemental composition (assigned into 5 major classes) are presented in Figure S6. The
applied rule-based classification scheme distinguishes among particle-types common in the remote marine environment of "Sea
Salt" , "Sea Salt/Sulfate", "Carbonaceous/Sulfate", "Carbonaceous", and "Other" (Figure S6). For each sample, 600–3000 particles
were analyzed, depending on particle loading on the substrates. The size-resolved particle-type classification identified using rule-
based schematic were overlaid on the acquire FIMS size distribution as shown in Figure S7. Similar to the k-means clustering
break down, we first compared the impact of DI events in MBL samples. Significant fractions of "Carbonaceous" and
"Carbonaceous/Sulfate" particles were identified in the background MBL samples amounting to 86% (non-DI periods) while
decreasing to 49% (DI period). Furthermore, the combined fraction of "Sea Salt" and mixed "Sea Salt/Sulfate" are substantially
smaller around 10% (non-DI periods) to 21% (DI period). Fractions of uncategorized "Other" particles contributes to around 30%
(DI period) while only having marginal influence of 4% during non-DI events. In contrast, background FT samples were dominated
by "Carbonaceous" and "Carbonaceous/Sulfate" contributing to as high as 95% (non-DI periods) then decreasing to 55% (DI
period). Unlike the MBL samples, there were only minimal change in larger "Sea Salt" + mixed "Sea Salt/Sulfate" from 2% (non-
DI period) to 4% (DI period). However, the reduction in "Carbonaceous" and "Carbonaceous/Sulfate" contribution among FT
samples during DI periods is due to associated with the large change in "Other" fraction shifting from 4% (non-DI period) to 41%
(DI period). Based on the mean elemental composition of the "Other" category, this group contains a combination of dust, sea salt,
and carbonaceous components suggesting extensive internal mixing of particles consistent with long-range transport (Froyd et al.,
2019). This finding is also consistent with the $k$-means clustering results that indicated elevated contributions of particles with
inorganic components during the DI periods. Overall, the particle-type fraction identified by both the $k$-means clustering and the
rule-based classification schemes are consistent across all samples suggesting that the mixing state population significantly changes
from heavily organic dominated to a mixture of inorganic-organic particle-type distribution resulting in the observation of more
complex particle compositions during DI periods.

Relative contributions of the particle-type fractions among separate DI events show substantial variability between different flights
and MBL versus FT altitudes (Figure S8). Furthermore, the dominant "Carbonaceous" particle-type groups identified by
CCSEM/EDX elemental analysis may exhibit significant differences in the spectral characteristics of carbon bonding indicative of
its long-range transport from North America during the DI periods. Furthermore, a previous study have tracked the origin of air
masses transported over long distances across the Atlantic Ocean to the Azores utilizing a Lagragian Flexible Particle
(FLEXPART) dispersion model to show detailed spatial resolution of air masses across different locations and altitudes (China et
al., 2017). The influence of North American emissions on distant remote regions is well documented with occurrences of
continental pollutant transport events accompanied by strong influence from urban city emissions spanning from Boston, Toronto,
Detroit, and Chicago (Owen et al., 2006). On the other hand, extensive boreal wildfires in northern North America release large
amounts of trace gases and aerosols into the atmosphere, which then can be transported to other remote regions including North
America (Val Martín et al., 2006). In particular, boreal wildfires emit around 10% of the annual anthropogenic aerosol black carbon





in the Northern Hemisphere (Bond et al., 2004). The eastward transport of North American emissions begins as hot plumes of

biomass burning emissions from wildfires rapidly rise to high altitudes (~8 to 13 km AGL) under favorable conditions (Zhu et al., 2018; Yu et al., 2019; Kloss et al., 2019). These plumes can be lofted into a warm conveyor belt preceeding a cold front from an associated cyclone, which is followed by the entrainment of a cold descending air stream (from the same cyclone) that ultimately results in the air parcels containing continental emissions reaching the lower altitudes of the Eastern North Atlantic (Owen et al., 2006; Zheng et al., 2020b). The transported aerosol undergoes substantial atmospheric aging through photochemical reactions

(Hems et al., 2021), gas-particle partitioning (Vakkari et al., 2018), and coagulation (Ramnarine et al., 2019) processes as it travels across the Atlantic ocean and descends into the MBL during the DI events.

### 3.3 Internal mixing of individual particles

Results of the elemental microanalysis of particles presented above provides statistics on broad particle classes identified and shows well the significant contribution of organic dominated particles in the region. However, CCSEM/EDX analysis is limited in

providing detailed information on the carbon speciation within individual particles and other metrics of particle internal composition (chemical mixing state). To investigate chemical differences in the carbon components of particles we employed STXM/NEXAFS spectro-microscopy methods, which provide spatially resolved carbon bonding speciation and differentiate between EC and OC regions within individual particles (Moffet et al., 2010a, c). Figure 3A shows an illustrative carbon K-edge map of individual particles from one of the DI period samples (the cumulative map of all ~4,300 particles from all samples analyzed

in this study is included in SI, Figure S9). The carbon K-edge composition map distinguishes 3 main components based on the spectral information (Moffet et al., 2010a) as described earlier: IN (blue), OC (green), and EC (red). Each pixel within an individual particle may contain either single or multiple components (i.e., components can overlap) that are grouped to yield 5 typical classes based on the internal mixing between OC, EC, and IN components: (1) IN, (2) OC-EC-IN, (3) OC-EC, (4) OC-IN, and (5) OC. The size-resolved histograms of these 5 classes superimposed with the onboard particle size distribution data measured by FIMS

is shown in Figure 3B to highlight the organic/inorganic contributions within individual particles as a function of particle size. A mixture of organic and inorganic particles (OC-IN) appears to be the dominant class across all samples, contributing 40–76% to the total particle population. Furthermore, the consideration of multiple sources of EC from wildfires (Park et al., 2007), residential wood smoke (Allen and Rector, 2020), agricultural burning (Liu et al., 2016; Holder et al., 2017), and urban emissions (Paredes-Miranda et al., 2013) in North America led us to expect large contribution of EC within our sample. However, OC-EC and OC-

EC-IN particles contributed only 0.4–1.3% to the total particle population. EC/soot lifetime is primarily governed by its wet deposition rate, which is dependent on the particle's affinity to absorb water (Barrett et al., 2019). Freshly emitted soot particles are hydrophobic, however atmospheric processes can increase the hydroscopicity properties of soot particles through the accumulation of OH initiated oxidation of organics during long-range transport and atmospheric aging (Dzepina et al., 2015) leading to decreased atmospheric lifetime of EC regardless of initial composition (Khalizov et al., 2013; Browne et al., 2015; China

et al., 2015). IN particles (i.e., inorganics such as sea salt and sulfates) appear to be consistent with the particle-type observations inferred from CCSEM/EDX data. Singe-component IN particles contribute up to 15% in the MBL at the time of no-DI periods, while their contribution during DI decreases to ~0.8%. Subject to long-range transport, IN dominant particles also accumulate substantial OC components encountering the DI, and as they entrain into the MBL and create ensembles of ambient particles with complex multi-component internal mixing states through different atmospheric processes such as condensation (Mozurkewich,

1986) and coagulation (Holmes, 2007). Consistently, fractions of single-component OC particles within the MBL during DI periods increased (from 7% to 22%) and slightly decreased in the FT layer (from 26% to 20%). These observations suggest that entrainment





of aerosols with higher extents of internal mixing (from long-range transport) are present in the MBL and can contribute to the regional aerosol composition, which in turn may modify aerosol-cloud interactions typical for the area.

NEXAFS spectra (285–294 eV) of individual particles were used to assess carbon chemical bonding environment allowing us to

identify representative types of OC containing particles (Moffet et al., 2010a). Figure 4 shows the representative NEXAFS spectra acquired over 103 individual carbon containing particles. This resulted in the identification of 6 carbon "types," as shown along with their illustrative SEM images. Each carbon "type" is classified based on characteristic spectral features such as peak positions and relative intensities. For all spectra shown in Figure 4A, the individual contribution of carbon energy transitions was quantified via spectral deconvolution. Details on the deconvolution process are described in previous works (Moffet et al., 2010b, 2013;

Tomlin et al., 2020). Figure 5A shows the deconvolution fit of the averaged NEXAFS spectra for each carbon "type" identified across different sampling conditions with Figure 5B illustrating the contribution of each functional group based on the individual peak area. It is worth noting that the difference in absorption between the post-edge ($OD_{320\,eV}$) and pre-edge ($OD_{278\,eV}$) energies is a measure of the amount of total carbonaceous material in the particles.

"Type 1 – biological" class has some contribution from alkene groups (C*=C @ 285.4 eV) with significant enhancement of

aliphatic hydrocarbons (C*–H @ 287.7 eV) and alcohol groups (C*–OH @ 289.5 eV). These spectra appear to be similar to the reported NEXAFS spectrum for phospholipids, a constituent of cell walls (Lawrence et al., 2003; Nováková et al., 2008). Lipid material is concentrated in the sea surface microlayer through the rupturing of phytoplankton cell membranes (i.e., cell lysis) (Wang et al., 2015b). A majority of lipid compounds produced by phytoplankton in seawater include glyceroglycolipids, phospholipids, and triacylglycerols containing significant amounts of aliphatic, and alcohol groups (Harwood and Guschina, 2009).

The transition of aliphatic-rich organic species into the aerosol phase is governed by the bursting of bubble films (Blanchard, 1989) enriched in lipid organic species found on the surface of seawater (Wang et al., 2015b). "Type 2 – homogeneous organic particles" have almost equivalent peak contributions from each reported functional group as shown in Figure 5B. The NEXAFS spectrum for type 2 is quantitatively similar to those reported for organic particles from anthropogenic emissions in urban areas of of Mexico City (Moffet et al., 2010b) and Central California (Moffet et al., 2013). As the aerosol plume is transported away from the source

of emission, organic mass increases while the fraction of C=C decreases (Doran et al., 2007; Kleinman et al., 2008; Moffet et al., 2010b). As a result, organic functional groups build up with particle age such as carboxylic acids, carbonyl, alcohol, and other carbon–oxygen functional groups. It has been suggested that formation of these homogeneous organic particles likely results from the accumulation growth of primary emitted particles as they traveled further way from their emission source (Moffet et al., 2010b). "Type 3 – soot" had the largest contribution of C*=C @ 285.4 eV spectral feature (42% of peak area contribution). Based on

reported literature, this spectrum is comparable with atmospheric particles collected during various field studies of biomass burning emissions (Hopkins et al., 2007). Interestingly, particles collected from aircraft measurements during the Aerosol Characterization Experiment in Asia (ACE-Asia) campaign (Maria et al., 2004) from emissions over mixed combustion sources had near identical % sp$^2$ value around 41% (Hopkins et al., 2007).

Field and laboratory studies showed that sea salt particles can react with atmospheric water-soluble organic acids leading to

chloride depletion within particles (Laskin et al., 2012; Wang et al., 2015a). Consistent with these previous studies, fresh sea salt typically has an intact rectangular inorganic core with a carbon outer shell arising from a thin layer of carboxylic acid coating as indicated by the peak for R(C*=O)OH @ 288.5 eV. Accordingly, "Type 4" is referred to as "fresh sea salt" in this work. In addition, the minor quantity of carbonaceous material in Type 4, as inferred from the small difference between the post- and pre-edge energies ($OD_{320\,eV} - OD_{278\,eV}$) apparent from Figure 5A, further supports the observation of freshly emitted sea salt particles. In

contrast, "Type 5 – aged sea salt/organics" are sea salt particles that have reacted with carboxylic acid components of organic





aerosol condensate which results in a substantial contribution of the R(C*=O)OH @ 288.5 eV peak while retaining the carbonate peak C*O$_3$ @ 290.4 eV. Of note, "Type 5 – aged sea salt/organics" contain significantly more carbon mass than "Type 4–fresh sea salt/organics," as indicated by their NEXAFS spectrum. Finally, "Type 6 – K dominated" class is identified based on the appearance of characteristic potassium peaks at 297.1 eV (K*$_{L2}$), and 299.7 eV (K*$_{L3}$) with a percent contribution of ~51% relative

to the total peak area. Potassium-salt particles are common markers of biomass burning smoke (Andreae, 1983; Li et al., 2003). Large fractions of KCl particles are commonly emitted from both flaming and smoldering fires, while atmospheric ageing can transform them into K$_2$SO$_4$ and KNO$_3$ through multi-phase acid displacement reactions similar to those of NaCl (Li et al., 2003). However, these K dominated particles can also be release as mixed secondary particles containing fractions of organic species, methylsulfonic acid, trimethylamine, SO$_4^{2-}$, NH$_4^+$, and K from potential biogenic sources in oceans (Willis et al., 2017).

**3.4 Organic volume fraction of individual mixed organic-inorganic particles**

Organic volume fraction (OVF) is a practical parameter to assess reactivity (Worsnop et al., 2002; Folkers et al., 2003) and hygroscopicity (Wang et al., 2008; Schill et al., 2015; Ruehl et al., 2016) of mixed inorganic–organic particles. Based on the STXM/NEXAFS measurements of individual particles, OVF is defined as a ratio of the optical thickness of the organic components ($t_{org}$) divided by the total optical thickness of the particle ($t_{org} + t_{inorg}$) (Moffet et al., 2010a; Pham et al., 2017; Fraund et al.,

2019). STXM images collected at the carbon K-edge were used to calculate the OVF. The values of absorbance at the pre-edge (278 eV) and the post edge (320 eV) energies are related to the inorganic mass and the sum of inorganic + organic mass, respectively. Assuming specific values for densities (ρ) and mass absorption coefficients (μ) for the organic and inorganic components, values of $t_{org}$ and $t_{inorg}$ can be determined, allowing OVF calculation (Fraund et al., 2019). For this study, we assumed the inorganic component of particles corresponds to (NH$_4$)$_2$SO$_4$ based on the particle elemental composition identified by

CCSEM/EDX analysis, while oxalic acid (C$_2$H$_2$O$_4$) is used as a proxy for the organic component. Oxalic acid was chosen to represent biomass burning (Yamasoe et al., 2000) and vehicular exhaust (Kawamura and Kaplan, 1987). Of note, based on previous reported studies, assumptions of chemically different organic components has minor effect on the resulting OVF values, while choice of the inorganic components resulted in a larger variation in the OVF calculations (Pham et al., 2017; Fraund et al., 2019). Here, we estimate the systematic error in OVF when assuming different inorganic-organic components, as shown in Table S2.

Assuming NaCl to be the inorganic component instead of (NH$_4$)$_2$SO$_4$ yields a difference of ~35%. On the other hand, assuming the organic component to be oxalic acid yields a ~5–30% difference in OVF when compared to other organics depending such as sucrose, adipic acid, and glucose.

Figure 6 shows representative chemical mixing state maps and OVF values of particles sampled during different atmospheric transport episodes during this study. Particles appear to have varying amount of organic coating for different sampling episodes as

shown in the OVF maps. The comparison of the OVF map and the carbon speciation map illustrates overlap between the two mapping schemes. Finally, histograms show particle fractions at varying OVF values during different atmospheric transport episodes. Layers of organics are seen encapsulating inorganic cores. As expected, background particles collected in the MBL show inorganic NaCl cores (as indicated by a rectangular core morphology) with modest organic coating (OVF <30%), consistent with a previous report (Chi et al., 2015). However, during the DI periods, the majority of particles have equal or greater fractions of

organic to inorganic components (40–60% OVF), while only a few particles exhibit core/shell morphology typical for background particles (i.e., non-DI periods). Furthermore, FT particles during non-DI periods have OVF <10%, when compared to FT samples during DI periods (10–20% OVF). In general, samples collected at the FT altitudes show reduced OVF values compared to the MBL samples regardless of the occurrence of DIs. Core-shell particle morphologies were also observed in FT sample, albeit not frequently (see Figure S9). FT samples were dominated by inorganic-organic particles in the size range of 0.20–0.25 μm, which





are likely mixed sulfate-organic particles based on the size-resolved particle-type datasets obtained from CCSEM/EDX analysis. A recent study conducted in central Oregon found that the organic mass fraction from FT samples were between 27–84% while sulfate mass fractions were ranging from 39–50% (Zhou et al., 2019). Based on these reported studies the elevated contributions of organic and sulfate in the FT may be attributed to the enrichment of organonitrates and organosulfate compounds originating from biogenic sources in the absence of wildfire influence. However, FT organic and sulfate aerosol mass is also known to be

associated with urban and biomass burning emissions (Bahreini, 2003; Dunlea et al., 2009; Roberts et al., 2010; Wang et al., 2021b). Studies in the northeast Pacific found that submicron aerosol mass was dominated by sulfate and organic components originating from aged Asian pollution plumes (Dunlea et al., 2009). FT organic and sulfate particles can then experience long-range transport and aging as the air parcels are carried across the Atlantic and descend into the MBL of the ENA site (China et al., 2017). To summarize, we observe enhancements in the OVF values of individual particles during the DI periods, quantified as

2.06±0.16 and 1.11±0.04 fold increase of OVF for the MBL and FT samples, respectively, assuming $(NH_4)_2SO_4$–oxalic acid components. The larger total OVF in the MBL (relative to FT samples) regardless of DI events is most likely due to additional contribution of marine organic sources within the boundary layer. The background organic concentration in the MBL is different than FT due to other sources of organics such as dissolved organic matter on the seawater surface (Doval et al., 2001; Miyazaki et al., 2018). The transport of organics from the ocean surface directly into the atmosphere is primary driven by turbulent winds

(O'Dowd et al., 2004; Prather et al., 2013) resulting in the enhancement in the background organic concentration in the MBL. Furthermore, the observed enhancements in OVF in the MBL during DI periods could be the result of organic-rich air parcels (originating from North America) descending from the FT into the MBL leading to changes in total organic concentration (Zheng et al., 2020b; Wang et al., 2021b). It is also worth mentioning that the definitions of "Carbonaceous" particles identified by CCSEM/EDX is somewhat different from OC particles defined by STXM/NEXAFS. The former corresponds to the distribution

of organics across a population of particles (i.e., external mixing) while the latter is related to the multi-component internal heterogeneity of individual particles (i.e., internal mixing).

**3.5 Evaluating CCN activity of mixed organic-inorganic particles**

CCN activity of individual particles is governed by both their size and chemical composition. In particular, condensation of organic carbon onto atmospheric inorganic particles can impact the efficiency at which particles of mixed organic-inorganic composition

can act as CCN and INP due to changes in particles' hygroscopicity and viscosity (Beydoun et al., 2017; Ovadnevaite et al., 2017; Altaf et al., 2018). To account for the effects of organics on aerosol hygroscopicity, we use κ-Köhler equation (Petters and Kreidenweis, 2007) to estimate the hygroscopicity parameter κ corresponding to mixed inorganic–organic particles:

$$\kappa = (1 - f_{org})\, \kappa_{inorg} + f_{org}\, \kappa_{org} \tag{2}$$

where, $f_{org}$ is the OVF values derived from the STXM data, $\kappa_{org} = 0.1$ is the hygroscopicity of the organic component, and $\kappa_{inorg} =$

$0.6$ is that of $(NH_4)_2SO_4$ (Petters and Kreidenweis, 2007). We derived κ values for different synoptic and atmospheric layer conditions using the size-resolved OVF ratio shown in Figure S10, and found that $\kappa_{MBL,\ DI} = 0.41$ and $\kappa_{FT,\ DI} = 0.33$ for DI periods, and $\kappa_{MBL,\ non\text{-}DI} = 0.48$ and $\kappa_{FT,\ non\text{-}DI} = 0.36$ for non-DI periods. The lower κ values under DI periods are consistent with enhancements in the organic contribution.

Using κ, we can calculate the critical size of a dry particle (Figure S11) that can be activated under the supersaturation of 0.14%

(setting of the CCN counter deployed on G-1) (Petters and Kreidenweis, 2007). The theoretical CCN number concentrations are then estimated by integrating the FIMS-measured aerosol size distributions above the critical dry particle diameter. Figure 7 shows the results for the OVF-based calculations of theoretical CCN concentrations compared to the onboard CCN measurements at


0.14% supersaturation. There is a general agreement between calculated and measured CCN concentration, but FT cases appear to have a better agreement. The uncertainty in the calculated CCN is due to the supersaturation fluctuation of the CCN counter (0.13–

0.15%), as shown in Figure S11. The large error bars in the measured CCN are a result of the variability of the measured CCN during different sampling periods. We also note that the exact value of $\kappa_{org}$ may play a role in affecting the CCN calculation. So, theoretical CCN concentrations were also calculated using $\kappa_{org} = 0$, and the results were compared against the measured CCN concentrations in Fig. 7b. However, the impact of this change of $\kappa_{org}$ does not significantly change the agreement between the calculated and measured CCN concentrations. This result shows that calculating the CCN concentration using OVF values derived

from the STXM data and the κ-Köhler theory can be a good estimate of the actual CCN concentrations.

**4 Conclusion**

Here, we presented detailed chemical imaging of individual atmospheric particles collected over the Azores during long-ranged transport events. Air mass back trajectory calculations suggest that air parcels in the ENA region can be traced from more than 4000 km away from North America within a span of 48-72 hours. During these long-range transport episodes, aerosols undergo

substantial changes in size, morphology, and chemical composition among others as they are carried across the Atlantic Ocean and descend from the FT into the MBL altitudes over the ENA region. Chemical composition of elements of individual particles (~36,400) were quantified using CCSEM/EDX while a subset of particles (~4,300) was analyzed using STXM/NEXAFS to determine the particle internal mixing state and organic spatial distribution. Based on CCSEM/EDX analysis, we observe a substantial contribution of "Carbonaceous" particles which are the dominant particle-type across all samples. The fraction of

externally mixed "Carbonaceous" particles decreases during the DI periods, compensated by the increase of "Ammonium Nitrate/Sulfate" fraction. The elevated contribution of atmospheric nitrate suggests influence from anthropogenic and biomass burning emissions (Reff et al., 2009). This observation is consistent with the DI periods suggesting air masses originating from North America descend from FT to MBL over the ENA region. Interestingly, there is also an increase in particle-type diversity in the FT during DI periods most likely due to significant mixing during DI episodes based on measured particle number

concentrations. Among these identified "Carbonaceous" particles, the OVF across individual particles derived from STXM measurements is enhanced in DI samples. Aged aerosols accumulate organics through condensation of secondary semi-volatile species resulting in an increase in organic contribution among individual particles. We utilize the STXM-derived OVF values and implemented it into the calculation of particle hygroscopicity using κ- Köhler theory (Petters and Kreidenweis, 2007). Particles collected during DI periods resulted in lower κ value with respect to background marine aerosols common in the ENA region

resulting in reduced CCN propensity. We calculated κ values between ~0.29 and ~0.44 corresponding to mixed organic-inorganic aerosol in the FT and MBL, respectively. These values are consistent with previous reported studies on mixed organic particles (Petters and Kreidenweis, 2007; Schmale et al., 2018; Zheng et al., 2020b).

Current atmospheric models lack the representation of aerosol mixing states limiting to only simple assumptions leading to high uncertainty of aerosol impact on the Earth's system. It is traditionally assumed that sulfate particles dictate particle growth over

remote ocean regions while underestimating the influence of organic particles on the CCN activity over remote oceans. We have shown that particles transported from North America can have a substantial impact on the aerosol mixing state and aerosol population over the region of study, as organic contribution and particle-type diversity is significantly enhanced during the DI periods. These observations need to be considered in current atmospheric models to have a better predictive understanding of the impact of long-range transport episodes to the source apportionment of specific aerosol particle types and the extent of particle

internal heterogeneity.



*Supplement*

The supplement related to this article is available online at:

*Author Contribution*

D.V., S.C., D.K., R.M., J. Wang and A.L. designed the study. D.V., J.S. and J. Wang executed sample collection and data
acquisition during field deployment. S.R.-R. performed modeling tasks of the study. J.T., K.J., D.V., S.C., P.W., M.F., J. Weis,
F.R.-A., D.K., R. C., M.G. performed chemical imaging experiments and analyzed associated data. G.Z., Y.W. and J. Wang
analyzed real-time data from G-1. J.T. and A.L. wrote the manuscript with contributions from all coauthors.

*ORCID*

Jay M. Tomlin: 0000-0002-3081-1512
Kevin A. Jankowski: 0000-0003-4640-5161
Daniel P. Veghte: 0000-0001-7422-7791
Swarup China: 0000-0001-7670-335X
Matthew Fraund: 0000-0002-7460-4283
Guangjie Zheng: 0000-0002-8103-2594
Yang Wang: 0000-0002-0543-0443
Felipe A. Rivera-Adorno: 0000-0002-7355-7999
Shira Raveh-Rubin: 0000-0001-6244-8693
John Shilling: 0000-0002-3728-0195
Daniel Knopf: 0000-0001-7732-3922
Ryan C. Moffet: 0000-0002-2352-5454
Mary K. Gilles: 0000-0002-0672-3117
Jian Wang: 0000-0002-2815-4170
Alexander Laskin: 0000-0002-7836-8417

*Competing Interests*

The authors declare that they have no conflict of interest.

*Acknowledgements*

We would like to thank the ACE-ENA campaign team for their help and support. The research funded by the Atmospheric System
Research (ASR) program, Office of Biological and Environmental Research (OBER) of the United States Department of Energy,
awards No. DE-SC0018948 (Purdue/STI group), DE-SC0020259 (Washington University), SC0016370 and SC0021034 (Knopf
group), SR-R acknowledges funding from the Israel Science Foundation (grant No. 1347/18). The research used STXM/NEXAFS
instruments at beamline 5.3.2.2 and 11.0.2 at the Advance Light Source at Lawrence Berkeley National Laboratory with the
guidance from David Kilcoyne, Matthew Markus, Hendrik Ohldag, and David Shapiro. In addition, the soft X-ray
spectromicroscopy 10ID-1 beamline at the Canadian Light Source was also used in this study, assisted by the beamline scientist
Jian Wang. We used CCSEM/EDX instrument at Environmental Molecular Sciences Laboratory located at the Pacific Northwest
National Laboratory. We thank John Shilling for providing the data collected from the Aerodyne HR-ToF-AMS onboard the G-1
aircraft during the ACE-ENA campaign. The authors gratefully acknowledge the NOAA Air Resources Laboratory (ARL) for the





provision of the HYSPLIT transport and dispersion model and/or READY website (https://www.ready.noaa.gov) used in this publication.

*Financial Support*

U.S. Department of Energy Atmospheric System Research program, award DE-SC0018948.



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

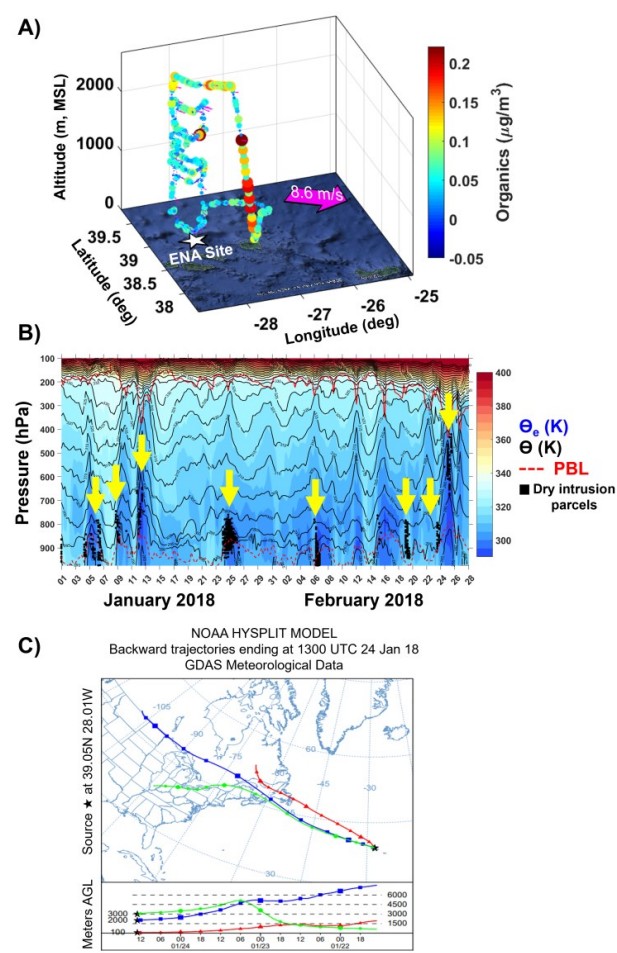

**Figure 1. A) A representative flight path of G-1 aircraft during one of the DI events (2018-01-24) during ACE-ENA campaign (Azores, Portugal). Size and color scale correspond to organic concentration provided by onboard Aerodyne HR-ToF-AMS. B) A time-height cross section at 39N,28W using ERA Interim reanalysis of ECMWF, showing equivalent potential temperature (K, shading), potential temperature (black contours) and boundary-layer height (red dashed line). The dolid red line is the 2-PVU contour of potential vorticity, marking the tropopause. The time periods of DI events (marked by black dots and indicated by yellow arrows) were identified from calculated forward trajectories based on the wind field data (ERA Interim, see text for more details). C) Calculated HYSPLIT 72 hrs back trajectory for 2018-01-24 utilizing GDAS1 arhived data sets starting at three elevations: 100 m (red), 2000 m (blue), 3000 m (green).**




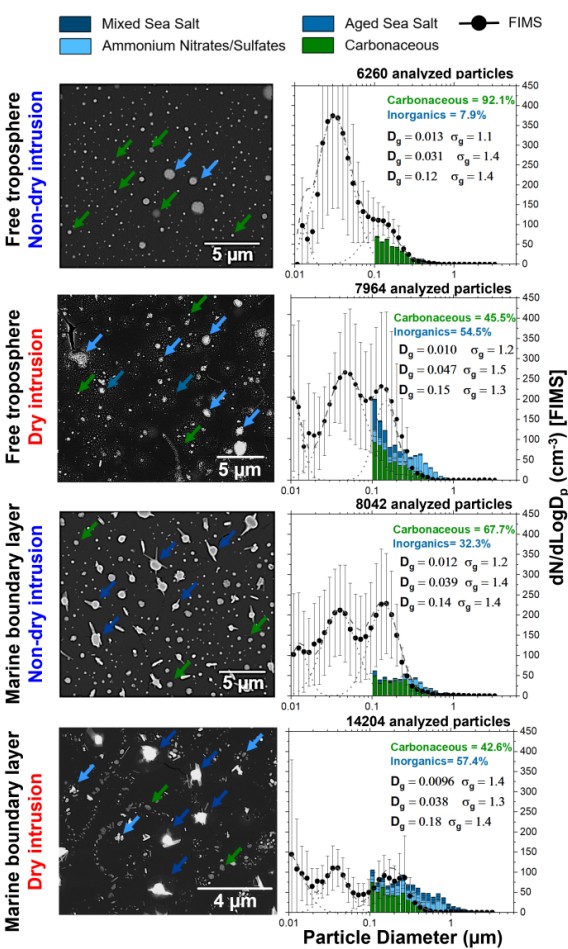

**Figure 2. Representative SEM images of particles (left column) and relative particle-type populations (right column) determined by CCSEM/EDX and *k*-means clustering analysis, summarized as a 16 bin/decade histogram representative of MBL and FT atmospheric layers and DI versus non-DI synoptic conditions. The composition of the size-segregated particle-type population were broken down into "Carbonaceous" and "Inorganics" (i.e., Mixed Sea Salt + Aged Sea Salt + Ammonium Nitrate/Sulfate). The average FIMS aerosol size distribution measured onboard G1 is superimposed and anchored at 0.25 µm to facilitate a visual assessment of particle types and number concentrations for CCN active particles (>100 nm). Lognormal mode diameter ($D_g$) and standard deviation ($\sigma_g$) were fitted for the FIMS particle size distribution (grey dashed lines).**





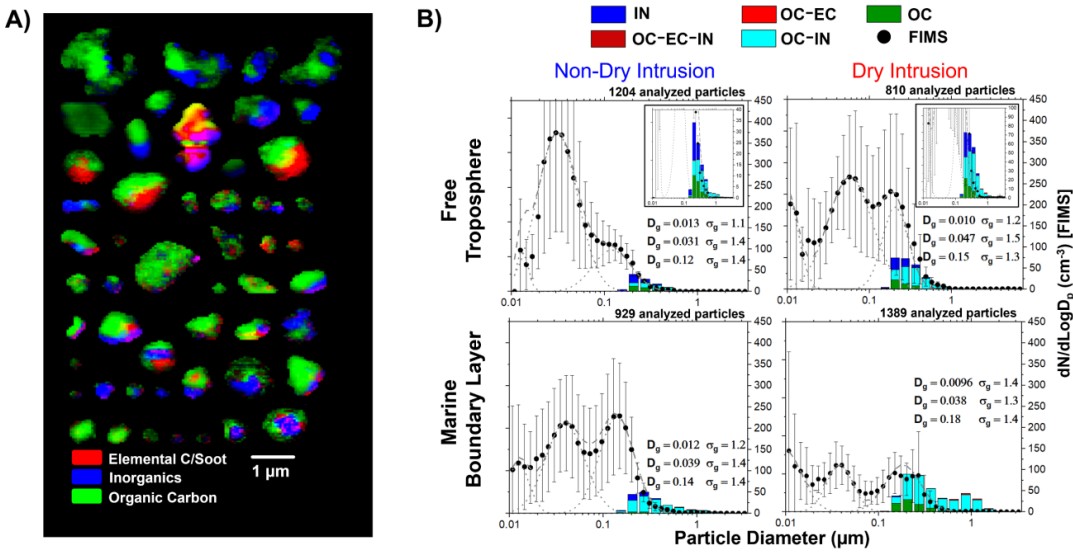

**Figure 3. A) Carbon speciation map of a subset of particles acquired by STXM from DI periods. Note that components can overlap where each pixel can contain different combination of the individual components: EC + IN constituents as purple; OC + EC as yellow; OC + IN as cyan. B) Size distribution of analyzed particles identified via STXM/NEXAFS shown as an 8 bin/decade histogram to compare particle multi-component internal mixing state between atmospheric transport events. FIMS particle size distribution is overlaid to facilitate a visual comparison from the same atmospheric episodes. Shown legends are as follows: IN–inorganics, OC–organic carbon (i.e., COOH), EC–elemental carbon (i.e., sp$^2$ C=C carbon).**

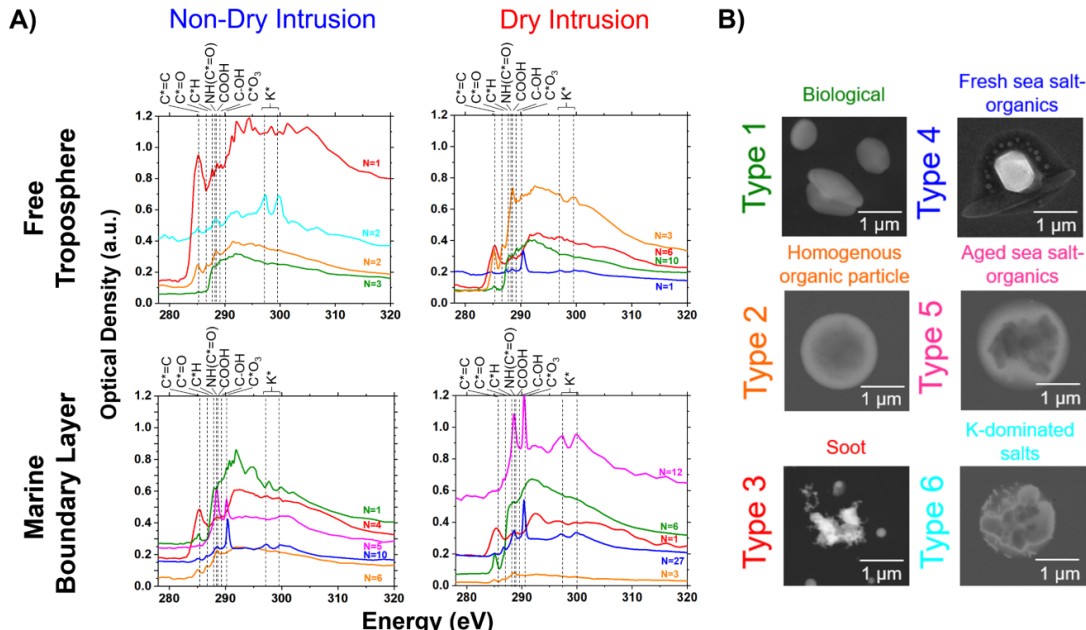

**Figure 4. A) Individual NEXAFS spectra showing differences in carbon content of representative particles collected at MBL and FT altitudes under different synoptic conditions. Identified Carbon types are: "Type 1–biological" (green), "Type 2–homogeneous organic particles" (orange), "Type 3–soot" (red), "Type 4–fresh sea salt/organics" (blue), "Type 5–aged sea salt/organics" (pink), "Type 6–K dominated salt" (teal). Dashed lines correspond to the transition energies: 285.4 eV (C*=C), 286.7 eV (C*=O), 287.7 eV (C*–H), 288.3 eV (R-NH(C*=O)R), 288.5 eV (R(C*=O)OH), 289.5 eV (RC*–OH), 290.0 eV (C edge step), 290.4 eV (C*O$_3$), 297.1 eV (K*$_{L2}$), and 299.7 eV (K*$_{L3}$). B) Representative SEM images of particles corresponding to the different carbon types identified with the STXM/NEXAFS analysis.**



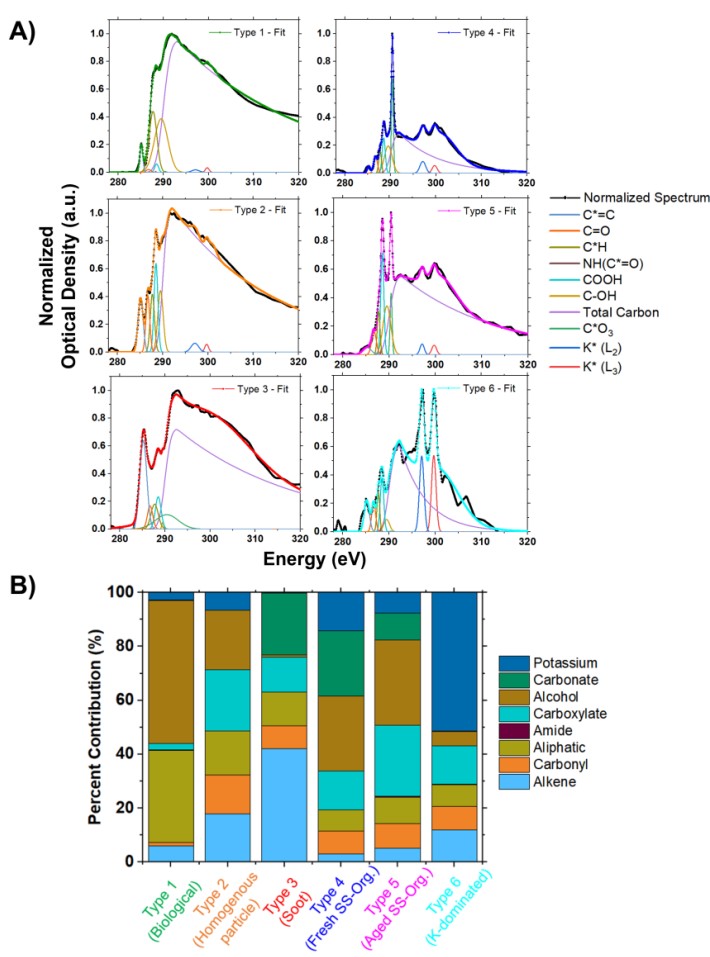

**Figure 5. A) Carbon K-edge NEXAFS spectra of 6 carbon types identified in individual particles: "Type 1 – biological" (green), "Type**
1055  **2 – homogeneous organic particles" (orange), "Type 3 – soot" (red), "Type 4 – fresh sea salt/organics" (blue), "Type 5 – aged sea salt/organics" (pink), "Type 6 – K dominated salt" (teal). B) Contributions of the different carbon functional groups reported as a percentage of the total peak area.**

1060



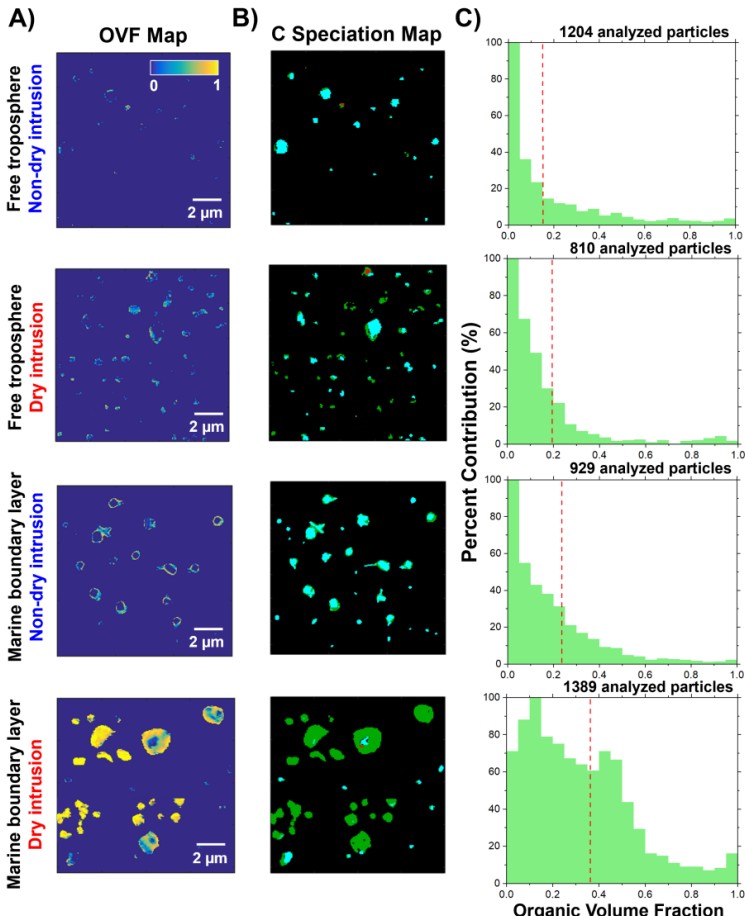

**Figure 6. A) Representative organic volume fraction (OVF) maps of individual particles. B) Carbon speciation maps of the identical particles; Teal – inorganic dominant regions; Green – organic dominant regions (i.e., COOH); Red – elemental carbon (i.e., sp² carbon). C) Histogram of particle fractions as a function of their OVF values with average OVF (red dashed line). The rows correspond to the different atmospheric layers and synoptic conditions to highlight the differences in organic/inorganic composition and multi-component internal mixing state of particles identified in this study.**

1065

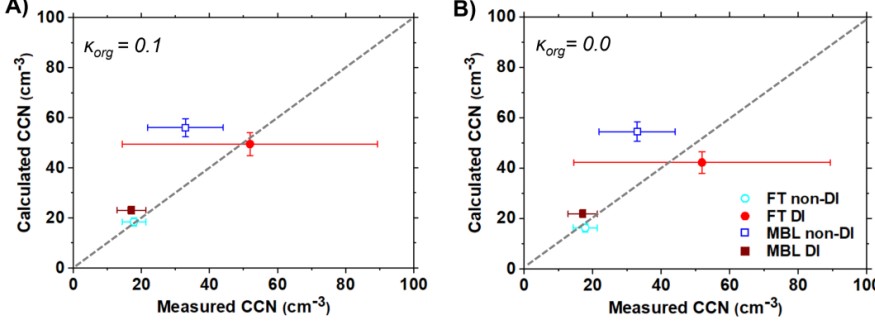

**Figure 7. Comparison of the CCN concentration predicted from the particle size distribution and OVF with field measured CCN by onboard instruments across the different atmospheric layer and transport event. A) $\kappa_{org}$ = 0.1; B) $\kappa_{org}$ = 0.0. Grey dashed line corresponds to the 1:1 calculated CCN to measured CCN.**