# Peer review of "Impact of Dry Intrusion Events on Composition and Mixing State of Particles During Winter ACE-ENA Study"

_Atmospheric Chemistry and Physics, 2021_

## Author Comment (AC1)

We thank the reviewers and the editor for careful reading and commenting on our paper. Our responses are marked below in blue, and edits made to the document are highlighted. A pdf file showing all edits to the original manuscript in tracking mode is also uploaded for editor's evaluation. All line numbers referenced in the rebuttals correspond to tracking mode document.

Reviewer(s)' Comments to Author:

Referee#1

General comments:

This manuscript "Impact of Dry Intrusion Events on Composition and Mixing State of Particles During Winter ACE-ENA Study" by Tomlin et al. measured aerosol particles collected during ACE-ENA study using CCSEM and STXM. They considered meteorology and back trajectories to understand the aging and sources of measured particles. The number of individual particle analyses is significant. The text is clear and well written. The conclusion is well supported by their observation. I have several suggestions regarding their figures and measurements, but overall, I support this paper to be published on ACP.

Specific comments:

1. (Page 4, Line 140-141) After each flight, sample discs were sealed and refrigerated prior to transport for off-line multi-modal microscopy analysis.

What temperature were the samples refrigerated? Could particles be exposed to high relative humidity (RH), i.e., above the deliquescence RH of sulfate or sea salt, in the refrigerator? We apologize for the confusion. There was a miscommunication of how the samples were handled and stored. After each flight, the collection disks were removed from the impactor, covered/O-ring sealed with a plate. There was no refrigerated storage or shipping of the samples. Below is the corrected description of how samples were stored prior to shipment (lines 142-145).

After each flight, sample discs were taken off the TRAC, plated, and hermetically sealed prior to transport. Once samples were received in the lab, the sample grids used for the analysis were removed from the sealed plate and transferred into TEM grid boxes stored at room temperature and dry conditions in a desiccator cabinet.

2. (Page 7, Line 231-234) While "Carbonaceous" is the dominant particle-type found across all samples, we also see that the elemental composition of each cluster (Figure S5) has common contributions from C and O elements, suggesting substantial carbon content in all existing particles, likely because of condensation of organic content and coagulation of particles.

What about the contribution from the substrate? Carbon substrate (which type?) could also emit C and O signals when the particles are thin. We agree and this is a valid point by both reviewers that background contribution of C (from Carbon Type-B film TEM substrates) makes C content in particles not quantifiable. As a result, we also removed our comment that the contribution of C and O suggests contribution of carbon in

sampled particles (originally lines 244-246). Our revised discussion is focused on reporting the sorting/classifying particles based on the elements present.

3. (Page 7, Line 266-267) The other cluster of "Aged Sea Salt" particles shows significant fractions of Na (~10%), but with substantially lower ratios of Cl/Na<0.1 which indicates chloride depletion (Figure S5) due to atmospheric aging.

In Fig. S5, "Aged sea salt" and "Mixed sea salt" cluster groups include a substantial amount of Al and Si. Based on Fig. S6B, these cluster groups could be a mixture of sea salt + other particle types (possibly dust particles from long-range transport (Line 292)). Although it is understandable to use two different classification methods, I suggest discussing the contribution of dust particles for the clustering groups.
As suggested, we included a notion of sea salt mixed with mineral dust attributed to the long-range transport (lines 294-295):

Additionally, both "Aged Sea Salt" and "Mixed Sea Salt" cluster groups include minor contributions of Al and Si indicative of possible mixing with mineral dust transported from the long-range continental sources.

4. (Page 12, Line 455) $\kappa_{inorg} = 0.6$ is that of (NH4)2SO4 (Petters and Kreidenweis, 2007).

I suggest considering sea salt here in addition to ammonium sulfate because both sea salt and sulfate have been discussed in the paper.
As suggested, we added a notion explaining effects of plausible contributions from original and aged sea salt on the calculated kappa values of the mixed organic/inorganic particles (lines 484-486):

Of note, the values of $\kappa$ obtained here using Eq. 2 needs to be considered as the low limit values, which might be somewhat higher considering possible contributions from more hygroscopic components of particles related to original and aged sea salt ($\kappa_{NaCl}$=1.3 and $\kappa_{Na_2SO_4}$=0.8).

5. (Page 16, references) Atmospheric Chem. Phys.,

"Atmospheric" should be "Atmos." throughout the reference list.
Corrected.

6. (Page 26) Figure 1

I cannot read the details (e.g., equivalent potential temperature values and black dots for DI events) in Figure 1B in the PDF file.
We increased font size of equivalent potential temperature values and made black dot DI events smaller to see the finer detail between points.

7. (Page 27) Figure 2.

7-1. I assume projected area equivalent diameter (AED) was used for the SEM measurements judging from Page 9 in SI, and it should be also specified in the main text.
We added a description of AED obtained from CCSEM/EDX analysis (lines 167-169).

During CCSEM/EDX analysis particle samples were systematically imaged and particles larger than 100 nm are recognized. Of note, the particle size reported from CCSEM/EDX analysis is defined as the area equivalent diameter (AED, µm), which is based on fitting a circle with an area equivalent to the particle's 2D projected image,.

7-2. When comparing AED with FIMS size, they will not be able to be directly comparable because of different particle size definitions. I assume that AED can be larger than FIMS size because of particle flattering on the substrate. For example, it looks that the peak sizes of AED in Figure 2 right (e.g., the second one with FT and DI) is larger than those of FIMS. It may be helpful to discuss the difference in the particle size measurements.
As suggested, we added a notion explaining limitations that need to be considered while comparing AED and FIMS sizes. (lines 237-240):

Also, comparison of AED and FIMS sizes needs to be considered with caution because particle flattering on the substrate which results in overestimated of AED sizes, compared to more realistic FIMS values. Here, the AED based particle distributions are scaled to match Y-axis of FIMS data and therefore to provide visual illustration of the chemical makeup of CCN particles.

7-3. Y-axes in the right panels for SEM particle sizes are missing (?).
We scaled the normalized size distribution obtained from CCSEM and anchored it to the 0.25um from the FIMS size distribution, and therefore, the normalized particle number distribution values from CCSEM histogram are arbitrary. As a result, we decided to not show the y-axes values in Fig. 2 and Fig. 3.

7-4. Are the SEM images backscattering images, secondary images, or others?
For Figure 2, all SEM images used are backscattering mode. Added "backscattering mode imaging" in line 230 for clarification. However, for Figure 4, all SEM images are secondary images. We added "secondary electron mode imaging" in line 372.

7-5. Colors between "Mixed sea salt" and "Aged sea salt" are difficult to be distinguished.
Colors of "Mixed Sea Salt", "Aged Sea Salt", and "Ammonium nitrate/sulfates" have been adjusted to facilitate better visualization.

8. (Page 29) Figure 4

I suggest adding color references in Figure 4A, similar to Figure 5.
Added legends for each carbon NEXAFS.

Reviewer(s)' Comments to Author:

Referee#2

General comments:

The paper reports the observation of atmospheric particles in the marine boundary layer and free troposphere onboard a research aircraft flown over the Azores during dry intrusion events. The physiochemical properties of atmospheric particles were investigated using CCSEM and STXM. The data provided valuable information for understanding the influence of long-range transported air masses on the remote marine atmosphere. The figures are presented properly, and the paper is well organized. Overall, I recommend this study to be accepted after the revision.

Specific comments:

1. Line 232-233, did the authors consider the interference of the carbon membrane of TEM grid on C element analyses?

We agree and this is a valid point by both reviewers that background contribution of C (from carbon *B*-film TEM substrates) makes C in particles not quantifiable. As a result, we also removed our comment that the contribution of C and O suggests contribution of carbon in sampled particles (originally lines 244-246). And we want to focus on reporting the sorting/classifying particles based on the elements present.

2. Section 3.2, the authors should provide the definition of these four clusters at first, then discuss their fractions.

We included the definition of the 4 cluster in the beginning of section 3.2 (lines 241-248):

The *k*-means  algorithm identified 4 key clusters and were  termed as: "Carbonaceous", "Ammonium Nitrates/Sulfates", "Mixed Sea Salt" and "Aged Sea Salt" based on the mean elemental contribution (Figure S5). Note that the element fraction values obtained from individual EDX spectra were filtered to remove  values less than 0.5% .  "Carbonaceous" is the dominant  type and represents majority of analyzed particles. It is defined based on the sole contributions of C- and O-  elements from the particle EDX spectra. The second most abundant cluster is the "Ammonium Nitrates/Sulfates", where the contribution of N, O, and S are greater than 1%. The "Aged Sea Salt" and "Mixed Sea Salt" clusters contain similar elemental signatures with the latter containing significant amounts of refractory elements typical for sea salt and mineral dust including Mg, Cl, K, Ca, Mn, and Fe.

3. Line 270, if "Aged Sea salt" is mostly driven by carboxylic acids, I do not agree that "Inorganics" is the sum of "Mixed Sea Salt" + "Aged Sea Salt" + "Ammonium Nitrate/Sulfate" (Line 240).

As suggested, we added a notion explaining operationally defined "Inorganic" particles and the fact that it may include sea salt particles aged by organic acids. (lines 260-261).

Here, we operationally define "Inorganics"  as the sum of "Mixed Sea Salt" (4%) + "Aged Sea Salt" (20%) + "Ammonium Nitrate/Sulfate" (33%), which in fact may also contain organic carboxylic acids as components of aged sea salt.

4. Line 286, what is "marginal influence"?

Changed to "minimal contribution" (line 308).

==Fractions of uncategorized "Other" particles contributes to around 30% (DI period) while only having  minimal contribution of 4% during non-DI events.==

5. Line 434-435, how did the authors acquire the OVF enhancements? OVF values in the DI events compared with non-DI events?

Yes, this is correct. We took integrated the normalized percent contribution as a function of OVF (see Figure 6C) and calculated the OVF enhancement values by comparing DI events relative to non-DI events.

6. In the CCSEM result, carbonaceous particles decreased during the DI periods, while the STXM result shows that OVF increased due to the DI event. Seems the results are self-contradictory. Or maybe I have some problem understanding the data. If the authors believe that "Carbonaceous" particles identified by CCSEM/EDX are external mixing and OC particles from STXM are internal mixing (Line 445), I would prefer they mention this earlier in the paper.

Thank you for the suggestion. We also see where the confusion can occur. We moved lines 451-454 that explains the differences in the interpretation of the data from CCSEM and STXM earlier in section 3.3 (lines 345-349). See below,

To investigate chemical differences in the carbon components of particles we employed STXM/NEXAFS spectro-microscopy methods, which provide spatially resolved carbon bonding speciation and differentiate between EC and OC regions within individual particles (Moffet et al., 2010a, c). ==It is also worth mentioning that the definitions of "Carbonaceous" particles identified by CCSEM/EDX and described in the previous section is somewhat different from OC particles defined by STXM/NEXAFS. The former corresponds to the distribution of organics across a population of particles (i.e., external mixing) while the latter is related to the multi-component internal heterogeneity of individual particles (i.e., internal mixing).== Figure 3A shows an illustrative carbon K-edge map of individual particles from one of the DI period samples (the cumulative map of all ~4,300 particles from all samples analyzed in this study is included in SI, Figure S9).

7. Figure 1B, it would be better if the authors add black contour and red solid line in the annotation. PBL should be PBL height.

Added potential temperature (black contour) and potential vorticity (red solid line) in the annotation. We added "height" in PBL annotation.

Minor comments:
8. Line 25, physiochemical should be physicochemical.

Corrected.

9. Line 28-31, the sentence is too long to follow.

We separated the sentence into two. See below (lines 28-32).

==Chemical and physical properties of individual particles were investigated using complementary capabilities of computer-controlled scanning electron microscopy and X-ray spectro-microscopy to probe particle external and internal mixing state characteristics . Furthermore, real-time measurements of aerosol size distribution, cloud condensation nuclei (CCN)==

==concentration, and back trajectory calculations were utilized to help bring into context the findings from off-line spectromicroscopy analysis.==

10. Line 56 Atmospheric particles exhibit complex internal heterogeneity in the troposphere (Ref). I might think that more references should be added here including American, higher level and East Asia and so on.

*Li et al.,, A conceptual framework for mixing structures in individual aerosol particles. J. Geophys. Res. 2016, 121, (22), 13,784-13,798.*

*Riemer et al.,.: Aerosol Mixing State: Measurements, Modeling, and Impacts, Rev. Geophys., 57, 187–249*

*Buseck and Posfai., Airborne minerals and related aerosol particles: Effects on climate and the environment. P. Natl. Acad. Sci. USA 1999, 96, (7), 3372-3379.*
Added suggested references.

11. Line 209, the degree symbol should be in front of N or W.
Corrected.

12. Line 979, dolid should be solid.
Corrected.

13. Line 982, arhived should be archived.
Corrected.